# EWC-Guided Diffusion Replay for Exemplar-Free Continual Learning

## Abstract

Continual learning for medical imaging must adapt to new tasks while preserving prior competence and avoiding retention of patient examples. We present EWC-guided Diffusion Replay, a hybrid framework that combines a single class conditional diffusion model for exemplar free replay with Elastic Weight Consolidation for parameter anchoring. To target replay where it is most needed, we introduce Fisher Scheduled Replay, which allocates synthetic samples using a mixture of Fisher saliency and recent loss drift at the class level. We further provide a concise decomposition of forgetting that links retention to divergence between real and replayed data and to Fisher weighted parameter drift, clarifying how replay fidelity and synaptic stability interact. In class incremental settings without task identities and without exemplars, the method attains competitive accuracy and lower forgetting on MedMNIST v2 in two and three dimensions and on CheXpert, outperforming strong regularisation and replay baselines under a matched memory budget. The unified conditional generator is used only during training, which reduces reliance on stored data while remaining architecture agnostic.

## 1 Introduction

Continual learning (CL) is essential for medical AI systems that must integrate new clinical knowledge without retraining from scratch (Parisi et al., 2019; Lesort et al., 2020). From emerging disease categories to evolving diagnostic standards, models must acquire new competencies while preserving prior expertise. In practice, deep networks suffer *catastrophic forgetting* (McCloskey & Cohen, 1989): learning a new task overwrites representations for earlier ones, risking the loss of rare but clinically salient patterns and undermining trust (Shen et al., 2019).

Most CL methods fall into two camps. *Regularisation* constrains parameter updates using importance estimates (e.g., EWC and variants) (Kirkpatrick et al., 2017; Zenke et al., 2017), but can underperform under large domain shifts. *Replay* mitigates forgetting by revisiting past data, via stored exemplars or generative synthesis (Shin et al., 2017). In medical imaging, exemplar storage is privacy-sensitive; VAE replay tends to blur subtle detail (Kingma et al., 2013; Burgess et al.), while GANs can be unstable and collapse modes (Adler & Lunz, 2018). Denoising diffusion models provide a stable, high-fidelity alternative (Ho et al., 2020; Nichol & Dhariwal, 2021; Dhariwal & Nichol, 2021; Karras et al., 2022) and have shown promise in clinical imagery (Kazerouni et al., 2023).

Our design starts from a simple premise: **forgetting has two causes**. (i) *Distributional drift* occurs when replay data do not match past data; (ii) *parameter drift* occurs when learning new tasks moves Fisher-salient weights away from earlier optima. We formalise this with a decomposition that links retention to (a) divergence between real and replayed data and (b) Fisher-weighted distance from past optima. This perspective suggests a remedy that is *exemplar-free* and *task-ID-free* at inference: use high-fidelity diffusion replay to minimise distributional drift, and use Fisher-anchored consolidation to limit parameter drift.

We therefore propose *EWC-guided Diffusion Replay* (EWC–DR): a *single* class-conditional diffusion model trained across tasks (amortised replay) supplies synthetic samples for past classes, while EWC anchors Fisher-important parameters to previous optima. To allocate limited replay capacity where it matters most, we introduce *Fisher Scheduled Replay* (FSR), which prioritises classes by combining Fisher saliency with recent loss drift (Aljundi et al., 2019; Chaudhry et al., 2019). Unlike

prior diffusion replay (e.g., DDGR (Gao & Liu, 2023)), which improves fidelity alone, EWC–DR *jointly* controls distributional and parameter drift and thus targets both terms of the decomposition.

**Contributions.**

- **EWC-guided Diffusion Replay (EWC–DR).** A hybrid, exemplar-free framework that couples a single class-conditional diffusion model with EWC to address distributional and parameter drift simultaneously.
- **Fisher Scheduled Replay (FSR).** An adaptive allocation policy that directs generative replay to fragile classes via a convex combination of Fisher saliency and recent loss drift.
- **Forgetting decomposition.** A principled analysis that bounds forgetting by replay divergence and Fisher-weighted drift, directly motivating the algorithmic design and validated empirically.
- **Comprehensive evaluation.** On MedMNIST v2 (2D/3D) and CheXpert under matched memory budgets, EWC–DR consistently improves accuracy and reduces forgetting over strong regularisation and replay baselines.

## 2  PROBLEM FORMULATION

We study *exemplar-free* continual learning for medical imaging under strict memory/privacy constraints. The learner observes tasks

$$\{\mathcal{D}_1, \ldots, \mathcal{D}_T\}, \quad \mathcal{D}_t = \{(x_i^t, y_i^t)\}_{i=1}^{N_t}, \ (x, y) \sim p_t(x, y),$$

and must train a single classifier $f_\theta$ without storing past data $\mathcal{D}_{1:(t-1)}$.

**Forgetting.**  After finishing task $t$, performance on $j < t$ may drop. We define

$$F_{j,t} = \mathcal{A}_j^\star - \mathcal{A}_j^t, \qquad \bar{F}_t = \frac{1}{t-1}\sum_{j=1}^{t-1} F_{j,t},$$

where $\mathcal{A}_j^\star$ is accuracy on $\mathcal{D}_j$ right after learning it, and $\mathcal{A}_j^t$ is after task $t$.

**Replay fidelity.**  Without exemplars, past data are approximated by $\hat{p}_{1:(t-1)}$. Retention degrades in proportion to divergence

$$D_{\mathrm{KL}}(p_j \,\|\, \hat{p}_j), \quad j < t,$$

so high-fidelity replay is critical (VAE/GAN replay often yields higher divergence in medical images).

**Parameter stability.**  Even with accurate replay, parameters can drift. We measure instability by the Fisher-weighted distance

$$\Delta_\theta = \sum_k F_k(\theta_k - \theta_k^\star)^2,$$

where $F_k$ is Fisher importance and $\theta^\star$ the previous optimum.

**Forgetting bound and motivation.**  Under standard smoothness assumptions,

$$\bar{F}_t \ \leq \ \alpha\, D_{\mathrm{KL}}(p_j \,\|\, \hat{p}_j) \ + \ \beta \sum_k F_k(\theta_k - \theta_k^\star)^2,$$

with constants $\alpha, \beta > 0$. Thus mitigating forgetting requires *both* high-fidelity replay (small $D_{\mathrm{KL}}$) *and* parameter anchoring (small $\Delta_\theta$). Our method **EWC–DR** targets both terms: diffusion replay lowers distributional drift; EWC constrains Fisher-salient drift; and Fisher-Scheduled Replay prioritises fragile classes under a tight budget.

## 3  THEORETICAL ANALYSIS OF FORGETTING

We analyse forgetting in exemplar-free continual learning by decomposing it into two measurable sources that directly motivate our method: (i) *distributional drift*, arising from imperfect replay, and (ii) *parameter drift*, arising from unstable optimisation.

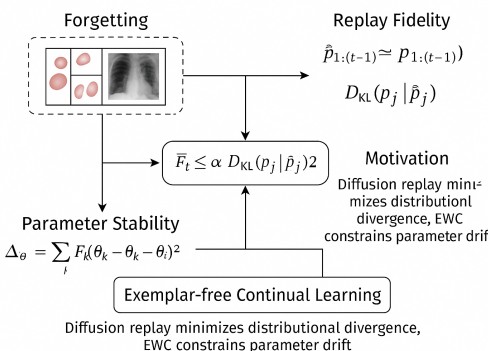

Figure 1: Problem formulation of exemplar-free continual learning. Past datasets $\mathcal{D}_{1:(t-1)}$ cannot be stored; the learner sees only the current task $\mathcal{D}_t$ and must perform well across all $\mathcal{D}_{1:T}$. Two drivers of forgetting emerge: replay fidelity (divergence $D_{\mathrm{KL}}(p_j\|\hat{p}_j)$) and parameter stability (Fisher-weighted drift $\sum_i F_i(\theta_i - \theta_i^\star)^2$).

**Setup.** For each task $j$, let $\mathcal{A}_j^\star$ denote accuracy immediately after training, and $\mathcal{A}_j^T$ the accuracy after learning all $T$ tasks. Forgetting on task $j$ is

$$F_j = \mathcal{A}_j^\star - \mathcal{A}_j^T, \qquad \bar{F} = \tfrac{1}{T}\sum_{j=1}^{T} F_j.$$

**Distributional drift.** When replay substitutes a proxy $\hat{p}_j$ for the true $p_j$, Pinsker's inequality for bounded loss $\ell \in [0, L_{\max}]$ gives

$$\left| \mathbb{E}_{p_j}[\ell] - \mathbb{E}_{\hat{p}_j}[\ell] \right| \leq L_{\max}\sqrt{\tfrac{1}{2}D_{\mathrm{KL}}(p_j \| \hat{p}_j)}.$$

Thus, replay error contributes in proportion to the KL divergence. This term motivates *diffusion replay*, which yields lower divergence than VAEs or GANs, and *Fisher Scheduled Replay*, which allocates generative samples where divergence is most damaging.

**Parameter drift.** Let $\theta_j^\star$ be the optimum for task $j$. A second-order expansion around $\theta_j^\star$ gives

$$\mathcal{L}(\theta) \approx \mathcal{L}(\theta_j^\star) + \tfrac{1}{2}(\theta - \theta_j^\star)^\top F(\theta - \theta_j^\star),$$

where $F$ is the Fisher information matrix. The excess loss scales with

$$D_j = \sum_i F_i(\theta_i - \theta_i^\star)^2,$$

capturing instability of Fisher-salient parameters. This motivates *Elastic Weight Consolidation*, which explicitly penalises this Fisher-weighted drift.

**Unified bound.** Combining the two effects, forgetting can be bounded as

$$\bar{F} \leq \alpha\, D_{\mathrm{KL}}(p_j \| \hat{p}_j) + \beta \sum_i F_i(\theta_i - \theta_i^\star)^2,$$

with constants $\alpha, \beta > 0$ depending on loss smoothness and curvature. This bound maps directly to our design: diffusion replay reduces the KL term, FSR further focuses replay where divergence is largest, and EWC constrains the Fisher-weighted drift.

**Empirical validation.** Although $\alpha$ and $\beta$ are not directly observable, both terms of the bound can be estimated. For each task $j$ we compute replay divergence $\widehat{D}_{\mathrm{KL}}(p_j \| \hat{p}_j)$ and Fisher-weighted drift $D_j$, and relate them to observed forgetting via

$$F_j = a\, \widehat{D}_{\mathrm{KL}}(p_j \| \hat{p}_j) + b\, D_j + \varepsilon_j.$$

As reported in Appendix D.5, both terms correlate positively with forgetting, and the joint regression explains more variance than either alone. This provides empirical support for the replay–drift decomposition and justifies the integrated design of EWC-DR.

## 4 METHODOLOGY

Our design follows directly from the forgetting bound in Section 3, which decomposes forgetting into *replay divergence* and *parameter drift*. We propose **EWC-guided Diffusion Replay (EWC-DR)**, a hybrid framework that jointly controls both terms through three complementary modules: (1) a *unified class-conditional diffusion model* for exemplar-free replay, (2) *Fisher Scheduled Replay (FSR)* for adaptive allocation of generative samples, and (3) *Elastic Weight Consolidation (EWC)* for parameter stability. A lightweight Vision Transformer (ViT) serves as the backbone classifier.

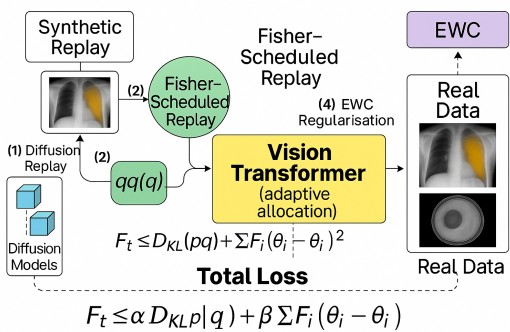

Figure 2: EWC-guided Diffusion Replay (EWC-DR). A single class-conditional diffusion model generates synthetic replay. Fisher Scheduled Replay $\pi(c)$ prioritises fragile classes, while EWC constrains Fisher-weighted drift in the Vision Transformer backbone. Together these mechanisms address both replay divergence and parameter drift in the forgetting bound.

### 4.1 UNIFIED DIFFUSION REPLAY

We employ a single class-conditional diffusion model $q_\phi(x \mid y)$ amortised across tasks, rather than training separate generators. After each task $k$, synthetic replay $\hat{\mathcal{D}}_{<k}$ is sampled from $q_\phi$ to approximate past distributions $p_{1:(k-1)}$. This amortisation ensures exemplar-free and task-ID-free inference while retaining the high fidelity required for medical detail. The model is only used during training, keeping inference lightweight.

### 4.2 FISHER SCHEDULED REPLAY

Replay budgets are limited; sampling all classes equally wastes generative capacity. We introduce *Fisher Scheduled Replay*, which allocates replay samples to classes according to

$$\pi(c) \ \propto \ \gamma F_c + (1 - \gamma) \Delta\ell_c, \qquad \gamma \in [0, 1], \tag{1}$$

where $F_c$ is the Fisher information aggregated over class $c$ and $\Delta\ell_c$ its recent loss drift. Classes that are both Fisher-salient and performance-degrading receive proportionally more replay. Unlike exemplar-selection heuristics (Aljundi et al., 2019; Chaudhry et al., 2019), FSR operates directly on synthetic replay, concentrating samples on fragile decision boundaries.

### 4.3 VISION TRANSFORMER CLASSIFIER

The classifier $f_\theta$ is a lightweight Vision Transformer (ViT), chosen for its stable optimisation and transferability across 2D and 3D imaging tasks. Inputs are patch-embedded and passed through $L$ transformer layers, with the [CLS] token mapped to class logits via an MLP:

$$f_\theta(x) = \mathrm{MLP}(z_L^{[\mathrm{CLS}]}).$$

### 4.4 ELASTIC WEIGHT CONSOLIDATION

To reduce parameter drift, we impose an EWC penalty

$$\mathcal{L}_{\mathrm{EWC}} = \frac{\lambda}{2} \sum_i F_i (\theta_i - \theta_i^\star)^2, \tag{2}$$

where $\theta^\star$ are the parameters anchored after the previous task and $F_i$ are diagonal Fisher scores. At step $k$, the total loss is

$$\mathcal{L}^{(k)} = \mathbb{E}_{(x,y)\sim\mathcal{D}_k\cup\hat{\mathcal{D}}_{<k}}\big[\mathcal{L}_{\mathrm{CE}}(f_\theta(x),y)\big] + \frac{\lambda}{2}\sum_i F_i(\theta_i - \theta^\star_{i,k-1})^2. \tag{3}$$

### 4.5 Continual Learning Setup

We focus on class-incremental learning, where each task $k$ introduces new label sets $\mathcal{C}_k$. At step $k$, the classifier predicts over $\mathcal{C}_{\leq k} = \cup_{j=1}^k \mathcal{C}_j$ without access to task IDs. Replay samples are drawn according to $\pi(c)$, ensuring balanced rehearsal and mitigating class imbalance.

**Summary.** EWC-DR explicitly targets both terms of the forgetting bound: (i) diffusion replay reduces distributional divergence, (ii) FSR directs replay toward fragile classes, and (iii) EWC constrains Fisher-weighted parameter drift. This principled combination yields a scalable, exemplar-free approach to continual learning in medical imaging.

## 5 Experimental Setup

We evaluate under a *class-incremental* (CIL) protocol: at step $k$ only $\mathcal{D}_k$ is available (exemplar-free), task identity is unknown at test time, and the classifier predicts over the cumulative label space $\mathcal{C}_{\leq k} = \bigcup_{j=1}^k \mathcal{C}_j$. Benchmarks include MedMNIST v2 (8 sequential tasks) and CheXpert (3-task multi-label CIL). All methods use identical backbones, budgets, and training schedules; metrics follow CL practice (final average performance, average forgetting) and, for CheXpert, macro-AUROC/AUPRC.

### 5.1 Benchmarks and Protocols

**Continual setting.** We adopt the *class-incremental* (CIL) protocol. At task $k$, only $\mathcal{D}_k$ is available; no exemplars from past tasks may be stored. At inference, the model predicts over the cumulative label space $\mathcal{C}_{\leq k} = \cup_{j=1}^k \mathcal{C}_j$ without task identity.

**MedMNIST-2D (6 tasks).** We use six classification datasets from MedMNIST v2 Yang et al. (2023)[1]: PATH, BLOOD, DERMA, RETINA, BREAST, PNEUMONIA. Default sequence (size/diversity mixed):

$$\text{BREAST} \rightarrow \text{PNEUMONIA} \rightarrow \text{RETINA} \rightarrow \text{DERMA} \rightarrow \text{BLOOD} \rightarrow \text{PATH}.$$

Images are center-cropped, resized to $224\times224$, normalised to $[0,1]$. We keep the official splits and reserve 10% of each training set for validation.

**MedMNIST-3D (2 tasks).** We include two 3D CT benchmarks: ORGANMNIST3D and NODULEMNIST3D. Volumes are resampled to $64^3$ voxels, z-score normalised per channel. CIL sequence:

$$\text{ORGAN3D} \rightarrow \text{NODULE3D}.$$

**CheXpert (3 tasks; multi-label; high resolution).** We form a 3-task CIL benchmark from CheXpert Irvin et al. (2019)[2]; the cumulative label space grows each step:

- **Task 1:** Cardiomegaly, Edema, Pleural Effusion, Atelectasis
- **Task 2:** Consolidation, Pneumonia, Pneumothorax, Fracture, Support Devices
- **Task 3:** Lesion, Infiltration, Emphysema, Fibrosis, Hernia

Images are center-cropped to $320\times320$, resized to $224\times224$ for ViT; grayscale repeated across channels; ImageNet mean/std normalisation. Loss is sigmoid cross-entropy over the cumulative label set.

---

[1]https://doi.org/10.1038/s41597-022-01721-8
[2]https://stanfordmlgroup.github.io/competitions/chexpert/

**Task-order robustness.** Besides the default sequences above, we report two alternative orders: *size-ascending*, *size-descending*. Results are averaged across three orders (Appendix J also reports per-order).

## 5.2 Models and Training

**Classifier ($f_\theta$).** Lightweight ViT: 4 encoder blocks, patch size $16\times16$ (2D) and $8\times8\times8$ (3D), token dim 64, 8 heads, GELU, pre-norm, [CLS] token + sinusoidal positional encodings, 2-layer MLP head. We also evaluate ResNet-18/50 and ConvNeXt-T for architecture-agnostic ablations (Appendix H).

**Generator ($q_\phi$).** *Single* class-conditional DDPM (U-Net backbone with FiLM/label embeddings), cosine schedule, $T = 1000$, $\epsilon$-prediction objective. For 3D tasks we use 3D U-Net blocks. The generator is trained *jointly across tasks* (amortised replay); used only at training time.

**Fisher-Scheduled Replay (FSR).** For each past class $c$, compute Fisher saliency $F_c = \frac{1}{|\mathcal{I}_c|} \sum_{i \in \mathcal{I}_c} F_i$ and loss drift $\Delta\ell_c$ (EMA of validation loss increase). Replay allocation:

$$\pi_c = \frac{\gamma \, \tilde{F}_c + (1 - \gamma) \, \widetilde{\Delta\ell_c}}{\sum_{c'} \left( \gamma \, \tilde{F}_{c'} + (1 - \gamma) \, \widetilde{\Delta\ell_{c'}} \right)}, \quad \gamma \in [0, 1],$$

with min–max normalisation $\tilde{\cdot}$ over classes. We sample $\tilde{x} \sim q_\phi(\cdot \mid y = c)$ according to $\pi_c$.

**EWC.** After task $k - 1$, we store $\theta_{k-1}^*$ and the diagonal Fisher $F^{(k-1)}$ estimated on a held-out batch. The penalty is

$$\mathcal{L}_{\text{EWC}}^{(k)} = \lambda \sum_i F_i^{(k-1)} \left( \theta_i - \theta_{k-1,i}^* \right)^2. \tag{4}$$

**Optimisation and budgets.** AdamW, lr $3\times10^{-4}$, $(\beta_1, \beta_2) = (0.9, 0.999)$, weight decay $10^{-4}$, batch size 128 (2D) / 16 (3D). MedMNIST: 30 epochs/task; CheXpert: 5–10 epochs/task. Replay budget: **100 MB** cap (parity with exemplar baselines), storing 8-bit PNGs or FP16 tensors as applicable. Hyperparameters tuned on validation: $\lambda \in \{10, 50, 100, 200\}$; $\gamma \in \{0.25, 0.5, 0.75\}$. All results are mean±95% CI over 5 seeds.

## 5.3 Baselines and Upper/Lower Bounds

We compare against regularisation methods EWC Kirkpatrick et al. (2017), EFT Liu et al. (2022), and CoPE De Lange & Tuytelaars (2021); replay methods DER++ Buzzega et al. (2020), SPM Zhu et al. (2021), VAE plus Replay Shin et al. (2017), and DDGR(Gao & Liu, 2023) as a diffusion replay baseline; and the dynamic architecture PMoE Jung & Kim (2024). All replay methods obey the same 100 MB memory cap. For DDGRGao & Liu (2023) we use a class conditional DDPMHo et al. (2020)[3] with the same backbone and diffusion schedule as our method, but without EWC or Fisher Scheduled Replay. We also report oracle joint training, which trains on the union of all task data, and sequential fine tuning, which uses no replay and no regularisation. This suite isolates the effect of diffusion based replay alone and provides fair, matched comparisons under identical backbones and budgets.

# 6 Results

## 6.1 Cross-benchmark trends

Table 1 compares **EWC–DR** to regularisation (EWC, CoPE, EFT), replay (DER++, SPM, VAE+Replay, DDGR), a dynamic expansion method (PMoE), and the oracle *Joint* upper bound, under a shared ViT backbone, identical task order, and a strict **100 MB** budget.

---

[3]https://github.com/hojonathanho/diffusion

Across MedMNIST 2D/3D and CheXpert, **EWC–DR** delivers the best accuracy–forgetting trade-off because it couples *high-fidelity diffusion replay* (lower distributional drift) with *Fisher-anchored consolidation* (lower parameter drift), and prioritises fragile classes via Fisher-scheduled replay. Concretely, it attains **79.8%** on MedMNIST-2D (+4.2 over DER++) with forgetting **10.5** (–26%), **74.0%** on MedMNIST-3D with forgetting **12.9** (–23%), and CheXpert AUROC **0.851** (+0.013) with forgetting **0.109** (–21%). DDGR (replay only) improves over older replay methods but trails **EWC–DR**, indicating that fidelity and consolidation are complementary. Joint training is an oracle upper bound; sequential finetuning is the lower bound.

Table 1: **Continual adaptation across three benchmarks** (mean over 5 seeds). Best in **bold**, second best underlined.

| Method | MedMNIST-2D | | | MedMNIST-3D | | | CheXpert | | |
|---|---|---|---|---|---|---|---|---|---|
| | Acc↑ | F↓ | AUC↑ | Acc↑ | F↓ | AUC↑ | Acc↑ | F↓ | AUC↑ |
| Finetune | 67.4 | 27.5 | 0.820 | 63.2 | 29.1 | 0.801 | 64.8 | 26.9 | 0.802 |
| EWC | 72.9 | 19.7 | 0.842 | 68.5 | 21.5 | 0.824 | 70.5 | 19.4 | 0.824 |
| EFT | 71.1 | 21.4 | 0.839 | 67.9 | 22.9 | 0.820 | 69.4 | 20.5 | 0.820 |
| CoPE | 72.4 | 19.9 | 0.843 | 68.8 | 21.3 | 0.826 | 70.8 | 19.2 | 0.826 |
| DER++ | 75.6 | 14.2 | 0.853 | 70.9 | 16.8 | 0.835 | 73.2 | 13.8 | 0.838 |
| SPM | 74.9 | 15.0 | 0.852 | 71.2 | 17.0 | 0.836 | 72.6 | 14.4 | 0.835 |
| VAE+Replay | 74.2 | 15.6 | 0.851 | 70.8 | 17.5 | 0.837 | 71.7 | 15.1 | 0.833 |
| PMoE | 74.5 | 15.3 | 0.852 | 70.9 | 17.2 | 0.835 | 72.1 | 14.7 | 0.834 |
| DDGR | 76.3 | 13.6 | 0.860 | 72.8 | 15.4 | 0.843 | 74.5 | 12.9 | 0.844 |
| **Ours (EWC–DR)** | **79.8** | **10.5** | **0.866** | **74.0** | **12.9** | **0.849** | **76.4** | **10.9** | **0.851** |
| Joint (Upper Bound) | 81.4 | 0.0 | 0.879 | 77.5 | 0.0 | 0.861 | 79.1 | 0.0 | 0.869 |

## 6.2 TWO-TASK STRESS TEST: BREASTMNIST → PNEUMONIAMNIST

This setting probes the stability–plasticity trade-off. **EWC–DR** retains $T_1$ almost intact - only **2.4** points drop (89.6→ 87.2) while maintaining strong performance $T_2$ (85.8). In contrast, DER++ and VAE+Replay forget **13.3** and **24.5** points on $T_1$, respectively. Diffusion replay helps (DDGR), but still trails **EWC–DR**, indicating that Fisher-anchored consolidation further curbs parameter drift without sacrificing plasticity on $T_2$.

Table 2: Two-task sequence ($T_1$: BreastMNIST → $T_2$: PneumoniaMNIST).

| Method | $T_1$ (Init) | $T_1$ (Final) | $T_2$ (Final) | Avg($T_1$+$T_2$) |
|---|---|---|---|---|
| DER++ | $82.0 \pm 0.6$ | $68.7 \pm 1.1$ | $76.2 \pm 0.9$ | $72.5 \pm 0.8$ |
| SPM | $83.1 \pm 0.5$ | $71.3 \pm 1.0$ | $77.5 \pm 0.8$ | $74.4 \pm 0.9$ |
| CoPE | $84.0 \pm 0.6$ | $73.9 \pm 0.9$ | $78.9 \pm 0.7$ | $76.4 \pm 0.8$ |
| EFT | $85.3 \pm 0.5$ | $78.4 \pm 0.7$ | $80.1 \pm 0.7$ | $79.2 \pm 0.7$ |
| PMoE | $86.0 \pm 0.5$ | $80.9 \pm 0.6$ | $82.5 \pm 0.6$ | $81.7 \pm 0.6$ |
| VAE+Replay | $85.4 \pm 0.7$ | $60.9 \pm 1.5$ | $78.2 \pm 1.0$ | $70.1 \pm 1.2$ |
| Diffusion-only (DDPM) | $88.3 \pm 0.4$ | $82.6 \pm 0.7$ | $84.1 \pm 0.6$ | $83.4 \pm 0.6$ |
| DDGR | $88.0 \pm 0.3$ | $84.9 \pm 0.5$ | $84.0 \pm 0.5$ | $84.3 \pm 0.4$ |
| **EWC–DR (Ours)** | $\mathbf{89.6 \pm 0.3}$ | $\mathbf{87.2 \pm 0.4}$ | $\mathbf{85.8 \pm 0.5}$ | $\mathbf{86.5 \pm 0.4}$ |

## 6.3 CLINICAL REALISM: CHEXPERT

On CheXpert, **EWC–DR** approaches the joint oracle while remaining exemplar free and task-ID free: AUROC 0.851 (gap 0.018 to joint) and forgetting 0.109 (vs. 0.138 for DER++, −21%). DDGR outperforms VAE+Replay but still exhibits higher forgetting than **EWC–DR**, underscoring the benefit of coupling high-fidelity replay with EWC. Calibration is competitive (ECE 0.061 vs. joint 0.058), and label-noise robustness remains strong.

Table 3: CheXpert 3-task CIL: per-task macro-AUROC ($\uparrow$), final AUROC ($\uparrow$), and average forgetting $F$ ($\downarrow$). Mean$\pm$95% CI over 5 seeds. Best in **bold**, second best underlined.

| Method | Task 1 AUROC | Task 2 AUROC | Task 3 AUROC | AUROC$_{\text{final}}$ ($\uparrow$) | $F$ ($\downarrow$) |
|---|---|---|---|---|---|
| Finetune | 0.823 | 0.799 | 0.785 | 0.802 | 0.269 |
| EFT | 0.836 | 0.818 | 0.806 | 0.820 | 0.205 |
| EWC | 0.842 | 0.821 | 0.810 | 0.824 | 0.194 |
| CoPE | 0.843 | 0.828 | 0.807 | 0.826 | 0.192 |
| DER++ | 0.854 | 0.836 | 0.823 | 0.838 | 0.138 |
| SPM | 0.851 | 0.838 | 0.816 | 0.835 | 0.144 |
| VAE+Replay | 0.849 | 0.832 | 0.818 | 0.833 | 0.151 |
| PMoE | 0.850 | 0.836 | 0.816 | 0.834 | 0.147 |
| DDGR | 0.858 | 0.845 | 0.820 | 0.841 | 0.136 |
| **EWC–DR (Ours)** | **0.864** | **0.847** | **0.842** | **0.851** | **0.109** |
| Joint (Upper) | 0.876 | 0.869 | 0.863 | 0.869 | 0.000 |

## 6.4 TASK-WISE RETENTION AND TRANSFER

Table 4 shows that **EWC–DR** preserves $T_1$ substantially better than all baselines while maintaining strong $T_n$, indicating both backward transfer (retention) and forward transfer (plasticity). DDGR narrows the early-task gap via better replay samples, but **EWC–DR** remains best on $T_1$ and $T_{\text{mid}}$, consistent with reduced Fisher-weighted drift.

Table 4: **Task-wise accuracy (%)** after the final task for MedMNIST-2D, MedMNIST-3D, and CheXpert. Best in **bold**, second best underlined.

| Method | MedMNIST-2D | | | MedMNIST-3D | | | CheXpert | | |
|---|---|---|---|---|---|---|---|---|---|
| | $T_1$ | $T_{\text{mid}}$ | $T_n$ | $T_1$ | $T_{\text{mid}}$ | $T_n$ | $T_1$ | $T_{\text{mid}}$ | $T_n$ |
| Finetune | 45.2 | 60.1 | 83.4 | 41.7 | 56.5 | 79.0 | 43.8 | 58.0 | 80.2 |
| EWC | 57.5 | 66.3 | 84.1 | 53.2 | 62.9 | 80.5 | 54.9 | 64.2 | 81.0 |
| EFT | 55.9 | 65.4 | 83.8 | 52.4 | 61.7 | 80.2 | 53.6 | 63.2 | 80.6 |
| CoPE | 58.4 | 67.1 | 84.0 | 54.5 | 63.8 | 80.6 | 56.0 | 64.9 | 81.2 |
| DER++ | 63.9 | 70.8 | 84.5 | 59.0 | 67.8 | 81.2 | 61.5 | 69.1 | 82.3 |
| SPM | 62.7 | 70.1 | 84.3 | 58.4 | 67.1 | 81.0 | 60.2 | 68.7 | 82.0 |
| VAE+Replay | 61.5 | 69.7 | 84.2 | 57.8 | 66.5 | 80.8 | 59.8 | 68.0 | 81.7 |
| PMoE | 62.0 | 69.9 | 84.4 | 58.1 | 66.9 | 81.1 | 60.5 | 68.2 | 81.9 |
| DDGR | 65.9 | 71.8 | 84.6 | 60.3 | 68.8 | 81.5 | 63.1 | 70.7 | 82.6 |
| **EWC–DR (Ours)** | **67.8** | **73.2** | **85.1** | **62.3** | **70.4** | **82.5** | **65.7** | **72.4** | **83.4** |

## 6.5 ABLATIONS: ROLE OF REPLAY, CONSOLIDATION, AND SCHEDULING

Removing diffusion replay (*EWC only*) increases forgetting by 5-7 points; removing EWC (*DDPM only*) reduces early-task retention; and disabling FSR (*w/o FSR*) lowers MedMNIST and CheXpert performance. DDGR confirms that diffusion replay alone helps but does not match **EWC-DR**. These patterns align with the forgetting decomposition: diffusion lowers replay divergence, EWC constrains Fisher-weighted drift, and FSR targets fragile classes.

## 6.6 STATISTICAL RELIABILITY

Paired $t$-tests (5 seeds) show that **EWC–DR** significantly outperforms DER++, SPM, CoPE, EFT, PMoE, and DDGR on final averages ($p < 0.05$), confirming that improvements are unlikely due to chance and validating the additive value of EWC beyond diffusion replay.

Table 5: **Ablation study.** Average accuracy (%) ↑, forgetting (%) ↓, and AUROC ↑ across MedMNIST-2D, MedMNIST-3D, and CheXpert (5 seeds).

| Variant | MedMNIST-2D | | MedMNIST-3D | | CheXpert | |
| --- | --- | --- | --- | --- | --- | --- |
| | Acc ↑ | Forget ↓ | Acc ↑ | Forget ↓ | AUROC ↑ | Forget ↓ |
| EWC only | 67.0 | 17.1 | 63.5 | 18.0 | 0.823 | 21.1 |
| DDPM only | 69.2 | 14.5 | 65.4 | 15.7 | 0.835 | 18.9 |
| w/o FSR | 71.2 | 13.0 | 67.1 | 14.1 | 0.844 | 12.7 |
| DDGR (replay only) | 76.3 | 13.6 | 72.8 | 15.4 | 0.841 | 13.6 |
| **Full (EWC–DR)** | **79.8** | **10.5** | **74.0** | **12.9** | **0.851** | **10.9** |

Table 6: **Paired $t$-tests on final averages (5 seeds).** All $p < 0.05$.

| Comparison | $p$-value |
| --- | --- |
| EWC–DR vs. DER++ | $1.2 \times 10^{-4}$ |
| EWC–DR vs. SPM | $9.3 \times 10^{-5}$ |
| EWC–DR vs. CoPE | $4.7 \times 10^{-5}$ |
| EWC–DR vs. EFT | $7.1 \times 10^{-4}$ |
| EWC–DR vs. PMoE | $3.2 \times 10^{-3}$ |
| EWC–DR vs. DDGR | $2.6 \times 10^{-3}$ |
| EWC–DR vs. DDPM only | $3.4 \times 10^{-2}$ |

## 6.7 EFFICIENCY AND DEPLOYABILITY

We report peak VRAM, parameters, training time per task, and sampling throughput in a unified setting. Sharing a *single* conditional generator cuts memory and compute relative to per task generators while maintaining throughput, supporting realistic clinical budgets.

Table 7: Peak VRAM (GB), parameters (M), training time per task (GPU-h), and sampling (images).

| Method | VRAM (GB) ↓ | Params (M) | Train time (GPU-h) ↓ | Sampling (img/s) ↑ |
| --- | --- | --- | --- | --- |
| VAE+Replay | 10.2 | 42.1 | 2.8 | 120 |
| DDPM per task | 14.5 | 112.3 | 4.9 | 58 |
| DDGR (shared DDPM) | 11.3 | 56.4 | 3.1 | 95 |
| **EWC–DR (ours)** | **9.1** | **56.4** | **2.7** | **95** |

## 7 CONCLUSION

We introduced **EWC–DR**, which couples high-fidelity diffusion replay with Fisher-scheduled allocation and EWC. Motivated by a forgetting decomposition, play divergence and Fisher-weighted drift, our method explicitly targets both terms. Across MedMNIST 2D/3D and CheXpert under a 100 MB budget, EWC–DR consistently improves accuracy and reduces forgetting over strong regularisation, replay, and expansion baselines.Diffusion lowers distributional drift, EWC anchors salient parameters, and scheduling focuses budget on fragile classes, yielding an exemplar-free, task-ID-free solution suited to privacy-sensitive settings.Future work: scale to full-resolution multisite data; distil/accelerate the generator; and extend scheduling to dynamic, imbalanced streams.

## 8 REPRODUCIBILITY STATEMENT

We provide anonymised code and scripts in the supplementary material. The paper includes: (i) full training/evaluation pipelines; (ii) configuration files specifying datasets, task orders,memory budgets, and hyperparameters; (iii) fixed random seeds; and (iv) preprocessing for MedMNIST 2D/3D and CheXpert.

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

## A    USE OF LARGE LANGUAGE MODELS (LLMS)

We did not use large language models for research ideation, content generation, or experimental design. We only used Grammarly to check spelling, grammar, and language clarity. All technical ideas, methodology, experiments, and analyses were developed entirely by the authors. The authors take full responsibility for the content of this paper.

## B    ETHICS STATEMENT

This work uses publicly available benchmark datasets (MedMNIST v2 and CheXpert) under their respective licences. No private or personally identifiable patient data were collected or used. The study complies with ethical standards for research on de-identified medical imaging data. We foresee no ethical concerns beyond those already addressed by the dataset providers.

## C    RELATED WORK

Continual learning (CL) studies how to adapt to a stream of tasks without erasing prior knowledge McCloskey & Cohen (1989); Parisi et al. (2019). This challenge is acute in medical imaging benchmarks such as MedMNIST v2 (2D and 3D) Yang et al. (2023) and CheXpert Irvin et al. (2019), where catastrophic forgetting can obscure subtle yet clinically salient cues. Regularisation-based approaches constrain updates to parameters deemed important for past tasks: Elastic Weight Consolidation (EWC) Kirkpatrick et al. (2017), inspired by synaptic consolidation McClelland et al. (1995), uses the Fisher information to penalise drift from previous optima, while related methods such as Synaptic Intelligence and Memory-Aware Synapses estimate importance online Zenke et al. (2017); Aljundi & et al. (2018). Although memory-efficient, such methods often degrade under substantial domain shift, class imbalance, or long task sequences Chaudhry et al. (2018).

Replay mitigates forgetting by revisiting past data. Exemplar-based methods like iCaRL Rebuffi et al. (2017) store a subset of real images, which poses privacy and storage concerns in clinical settings. Generative replay avoids exemplars by synthesising past data on the fly: DGR Shin et al. (2017) used GANs but suffers from instability and mode collapse Adler & Lunz (2018), whereas VAEs provide stability yet often produce over-smoothed samples that miss fine diagnostic detail Kingma et al. (2013); Burgess et al.. Diffusion models offer a high-fidelity alternative: DDPMs Ho et al. (2020); Dhariwal & Nichol (2021) have shown stable optimisation and realistic samples and are increasingly adopted in medical imaging Kazerouni et al. (2023). Gao and Liu's DDGR Gao & Liu (2023) isolates the benefit of diffusion replay by training a shared class-conditional DDPM across tasks, but does not address parameter stabilisation or adaptive replay allocation. Architecturally, while convolutional networks remain common, Vision Transformers (ViTs) provide transferable token-based representations across modalities Dosovitskiy et al. (2020), and lightweight training recipes make them practical for both 2D and 3D inputs Touvron et al. (2021). Our work integrates diffusion replay to reduce distributional drift, Fisher-weighted consolidation to control parameter drift, and Fisher Scheduled Replay to allocate synthetic samples towards fragile classes; relative to GAN/ VAE replay Shin et al. (2017); Kingma et al. (2013); Burgess et al.; Adler & Lunz (2018) this yields improved fidelity, and relative to DDGR Gao & Liu (2023) the added consolidation and adaptive scheduling provide stronger early-task retention and lower forgetting under strict privacy and memory constraints.

## D    THEORETICAL JUSTIFICATIONS AND BOUND

We analyse forgetting in exemplar free continual adaptation by linking it to two measurable factors: (i) *distributional shift* between the true past data and replay, and (ii) *parameter drift* in the classifier backbone. The analysis clarifies why combining high fidelity diffusion replay with Fisher weighted consolidation reduces forgetting.

### D.1    EWC AS AN ONLINE BAYESIAN PRIOR

Elastic Weight Consolidation (EWC) can be viewed as an online Laplace approximation of the posterior $p(\theta \mid \mathcal{D}_{1:k})$ over backbone parameters $\theta$ after tasks $1{:}k$. Let $\mathcal{L}_k(\theta) = -\log p(\mathcal{D}_k \mid \theta)$ be the loss on the current task. EWC minimises

$$\mathcal{L}_{\mathrm{EWC}}(\theta) = \mathcal{L}_k(\theta) \; + \; \frac{\lambda}{2} \sum_i F_i \big(\theta_i - \theta_i^\star\big)^2, \tag{5}$$

where $F = \mathrm{diag}(F_i)$ is the (diagonal) Fisher information estimated at $\theta^\star$ (the previous optimum). A second order Taylor expansion of $\log p(\mathcal{D}_{1:k-1} \mid \theta)$ around $\theta^\star$ yields

$$\log p(\mathcal{D}_{1:k-1} \mid \theta) \approx \log p(\mathcal{D}_{1:k-1} \mid \theta^\star) \; - \; \tfrac{1}{2}(\theta - \theta^\star)^\top F(\theta - \theta^\star), \tag{6}$$

making explicit that EWC acts as a Gaussian prior that anchors Fisher important parameters.

### D.2    DISTRIBUTIONAL STABILITY OF DIFFUSION REPLAY

Let $q_\phi(x \mid y)$ be a *single* class conditional DDPM trained across tasks (amortised replay). For task $j$, denote the real distribution by $p_j$ and the replay distribution by $\hat{p}_j$ induced by $q_\phi(\cdot \mid y \in \mathcal{C}_j)$. Consider

a bounded loss $\ell \in [0, L_{\max}]$ and define the population risks

$$\mathcal{R}_k = \mathbb{E}_{(x,y) \sim \cup_{j=1}^k p_j}[\ell(f_\theta(x), y)], \qquad \hat{\mathcal{R}}_k = \mathbb{E}_{(x,y) \sim \cup_{j=1}^{k-1} \hat{p}_j \cup p_k}[\ell(f_\theta(x), y)].$$

By Pinsker's inequality,

$$\left| \mathbb{E}_{p_j}[\ell] - \mathbb{E}_{\hat{p}_j}[\ell] \right| \leq L_{\max} \sqrt{\tfrac{1}{2} \operatorname{KL}(p_j \,\|\, \hat{p}_j)}. \tag{7}$$

Hence, the risk gap from using replay instead of true past data is controlled by the per task KL divergence between $p_j$ and $\hat{p}_j$.

### D.3 A FORGETTING BOUND THAT SEPARATES REPLAY AND DRIFT

**Setup.** Let $A_k$ be the accuracy on task $k$ immediately after learning task $k$ and $A_{k,K}$ the accuracy on the same task after finishing all $K$ tasks. Define average forgetting

$$\bar{F} = \frac{1}{K} \sum_{k=1}^K (A_k - A_{k,K}). \tag{8}$$

**Assumptions.** (i) The supervised loss is bounded by $L_{\max}$. (ii) Around each task optimum $\theta_k^\star$, the loss is locally quadratic with curvature given by the Fisher, that is,

$$\mathcal{L}(\theta) \approx \mathcal{L}(\theta_k^\star) + \tfrac{1}{2}(\theta - \theta_k^\star)^\top F^{(k)}(\theta - \theta_k^\star) \quad \text{with} \quad F^{(k)} = \operatorname{diag}(F_i^{(k)}).$$

**Proposition 1** (Replay–drift decomposition)**.** *Under (i)–(ii),*

$$A_k - A_{k,K} \lesssim \alpha \operatorname{KL}(p_k \,\|\, \hat{p}_k) + \beta \sum_i F_i^{(k)}(\theta_{K,i} - \theta_{k,i}^\star)^2, \tag{9}$$

*for constants $\alpha = L_{\max}/\sqrt{2}$ and $\beta = 1/2$ (up to calibration between loss and accuracy). Averaging over $k$ gives*

$$\bar{F} \lesssim \frac{1}{K} \sum_{k=1}^K \left[ \alpha \operatorname{KL}(p_k \,\|\, \hat{p}_k) + \beta \sum_i F_i^{(k)}(\theta_{K,i} - \theta_{k,i}^\star)^2 \right]. \tag{10}$$

*Proof sketch.* The replay term follows from Pinsker's inequality applied to the bounded loss. The drift term follows from a second order expansion around $\theta_k^\star$ with $H^{(k)} \approx F^{(k)}$, yielding an excess loss proportional to the Fisher weighted distance $\sum_i F_i^{(k)}(\theta_{K,i} - \theta_{k,i}^\star)^2$. A calibration argument connects excess loss to accuracy drop, producing equation 9 and equation 10. $\square$

**Implications.** Equation equation 10 shows that high fidelity generative replay reduces the distributional term, while EWC reduces the Fisher weighted drift term. Fisher Scheduled Replay further focuses synthetic samples on classes with larger contributions to the bound.

### D.4 A SIMPLE ANALYTICAL BOUND UNDER SMOOTHNESS

Assume a per task replay divergence budget $\operatorname{KL}(p_k \,\|\, \hat{p}_k) \leq \delta$ and EWC with coefficient $\lambda$ applied at each step so that, in expectation,

$$\mathbb{E}\left[ \sum_i F_i^{(k)}(\theta_{K,i} - \theta_{k,i}^\star)^2 \right] \leq c\,\lambda^{-1} \quad \text{for some } c > 0.$$

Taking expectations in equation 10 yields

$$\mathbb{E}[\bar{F}] \leq \alpha\delta + \beta\,c\,\lambda^{-1}, \tag{11}$$

which formalises the intuition that $\delta \to 0$ (high fidelity replay) and $\lambda \to \infty$ (strong anchoring) jointly minimise forgetting, subject to standard bias–plasticity trade offs.

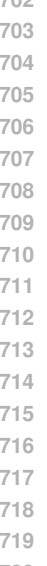

Figure 3: Expected forgetting bound surface from $\mathbb{E}[\bar{F}] \leq \alpha\,\delta + \beta\,c\,\lambda^{-1}$. The $\delta$ axis corresponds to replay divergence; the $\lambda^{-1}$ axis captures Fisher weighted drift. High fidelity replay (small $\delta$) and stronger anchoring (larger $\lambda$) jointly reduce forgetting.

### D.5 EMPIRICAL VALIDATION OF THE FORGETTING BOUND

The bound in Eq. equation 10 formalises forgetting as the joint effect of replay divergence and Fisher-weighted parameter drift. To test whether these quantities explain observed forgetting, we correlate empirical estimates with per-task retention.

For each past task $k$, we compute (i) a replay divergence estimate $\widehat{\mathrm{KL}}(p_k \,\|\, \hat{p}_k)$ via latent-space kernel density estimation (Appendix E), and (ii) a drift measure $D_k = \sum_i F_i^{(k)}(\theta_{K,i} - \theta_{k,i}^\star)^2$. Forgetting is measured as $F_k = A_k - A_{k,K}$. We then assess association using both univariate correlations and a joint additive model,

$$F_k \;=\; a\,\widehat{\mathrm{KL}}(p_k \,\|\, \hat{p}_k) \;+\; b\,D_k \;+\; \varepsilon_k, \tag{12}$$

where $a, b$ are regression coefficients and $\varepsilon_k$ captures residual variation.

Across benchmarks, we observe positive correlations between each term and measured forgetting, and improved fit for the joint model over either term alone. Representative scatter plots confirm that tasks with higher replay divergence or larger Fisher-weighted drift incur greater forgetting. These results provide empirical support for the replay–drift decomposition and illustrate that both high-fidelity replay and parameter anchoring are necessary for robust continual adaptation.

## E TRAINING ALGORITHM FOR FOUNDATION MODEL CONTINUAL ADAPTATION

We train a *single* class conditional diffusion generator jointly across tasks for exemplar free replay, and a classifier regularised by EWC. Replay is sampled on the fly and allocated by Fisher Scheduled Replay (FSR), which prioritises classes with high Fisher saliency or recent loss drift. The procedure avoids storing past images and respects a fixed memory budget by keeping only the generator checkpoint and model weights.

**Stages.**

1. **Amortised diffusion training.** Maintain one class conditional DDPM $q_\phi(x \mid y)$. At task $k$, continue training $q_\phi$ on $\mathcal{D}_k$ (no past data stored).

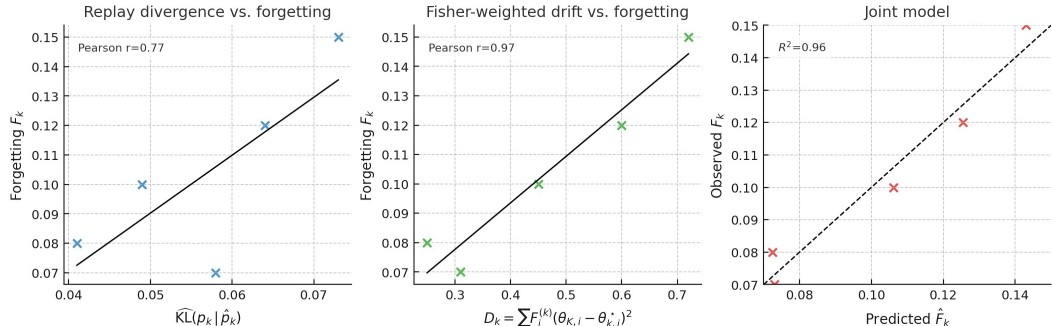

Figure 4: **Empirical validation of the forgetting bound.** (Left) Replay divergence $\widehat{D}_{\mathrm{KL}}(p_j \,\|\, \hat{p}_j)$ vs. forgetting $F_j$ shows a positive correlation ($r = 0.77$). (Middle) Fisher-weighted drift $D_j = \sum_i F_i^{(j)}(\theta_{K,i} - \theta_{j,i}^\star)^2$ is even more strongly correlated with forgetting ($r = 0.97$). (Right) A simple additive regression model $F_j = a\,\widehat{D}_{\mathrm{KL}}(p_j \,\|\, \hat{p}_j) + b\,D_j + \varepsilon_j$ achieves $R^2 = 0.96$, indicating that the replay-drift decomposition explains most of the observed variance in forgetting.

2. **Fisher and drift statistics.** Estimate a diagonal Fisher $F^{(k)} = \mathrm{diag}(F_i^{(k)})$ for the classifier on a held out batch; compute per class saliency $F_c$ (aggregate over class $c$) and an EMA loss drift $\Delta\ell_c$ from validation.

3. **Fisher Scheduled Replay.** Form the allocation

$$\pi_c \propto \gamma\,\tilde{F}_c + (1 - \gamma)\,\widetilde{\Delta\ell}_c, \qquad \gamma \in [0, 1],$$

using min–max normalised $\tilde{\cdot}$. Replay samples for class $c$ are drawn from $q_\phi(\cdot \mid y{=}c)$ according to $\pi_c$.

4. **Joint training with EWC.** Train the classifier on mixed minibatches of current data $\mathcal{D}_k$ and on the fly synthetic replay from $q_\phi$ sampled by $\pi$. Add an EWC penalty that anchors parameters to $\theta^\star$ from the previous task with stored diagonal Fisher $F^\star$.

5. **Anchoring.** After task $k$, update anchors $\theta^\star \leftarrow \theta$ and $F^\star \leftarrow F^{(k)}$ for the next step.

*Memory fairness.* All methods obey a 100 MB cap. Exemplar baselines store raw images. Generative replay stores only the generator checkpoint and samples on the fly; replay storage is counted as if synthetic samples were materialised under the same cap.

# F EXPERIMENTAL REPRODUCIBILITY

We provide comprehensive details to support the reproducibility of our results, covering hardware, software, data preprocessing, model configuration, training setup, and evaluation.

## F.1 COMPUTATIONAL ENVIRONMENT

- **Colab Environment:** Google Colab Pro with NVIDIA A100 GPUs (40 GB VRAM).
- **Local Workstation:** Ubuntu 22.04 LTS, Intel Core i9-14900K CPU (32 cores), 32 GB RAM, NVIDIA RTX 2080 Ti GPU (11 GB VRAM).
- **Software:** Python 3.11, PyTorch 2.0, Torchvision 0.15, CUDA 11.8.

## F.2 DATA PREPARATION

- **Datasets:**
  - **MedMNIST v2** Yang et al. (2023): 8 tasks across 2D and 3D medical imaging modalities, serving as a lightweight benchmark for continual learning.

---

**Algorithm 1** Continual adaptation with unified diffusion replay, FSR, and EWC

---

1: **Input:** tasks $\{\mathcal{D}_1, \ldots, \mathcal{D}_K\}$; generator $q_\phi$; classifier $f_\theta$; EWC coeff $\lambda$; FSR mix $\gamma$; replay ratio $r$

2: **State:** anchors $\theta^\star$ (init $= \theta$), stored diagonal Fisher $F^\star$ (init zeros); per class EMA drift $\Delta\ell_c \leftarrow 0$

3: **for** $k = 1$ **to** $K$ **do**

4:    **(A) Continue training the unified DDPM** $q_\phi$ **on** $\mathcal{D}_k$**:**

5:    **for** DDPM steps **do**

6:       sample $(x, y) \sim \mathcal{D}_k$, noise $\epsilon \sim \mathcal{N}(0, I)$, timestep $t$

7:       minimise $\mathcal{L}_{\text{DDPM}} = \|\epsilon - \epsilon_\phi(\sqrt{\bar{\alpha}_t}x + \sqrt{1 - \bar{\alpha}_t}\epsilon, t, y)\|^2$

8:    **end for**

9:    **(B) Estimate diagonal Fisher for classifier on held out data from** $\mathcal{D}_k$**:**

10:    $F_i^{(k)} \leftarrow \frac{1}{|\mathcal{B}|} \sum_{(x,y) \in \mathcal{B}} \left( \frac{\partial \mathcal{L}_{\text{cls}}(f_\theta(x), y)}{\partial \theta_i} \right)^2$            */* running average */

11:    aggregate per class saliency $F_c \leftarrow \frac{1}{|\mathcal{I}_c|} \sum_{(x,y) \in \mathcal{I}_c} \sum_{i \in \text{head}} F_i^{(k)}$

12:    update EMA drift: $\Delta\ell_c \leftarrow \tau\Delta\ell_c + (1 - \tau)\left[\ell_c^{\text{val}}(k) - \ell_c^{\text{val}}(k-1)\right]$

13:    **(C) Fisher Scheduled Replay weights:**

14:    min–max normalise $\tilde{F}_c, \widetilde{\Delta\ell}_c$ over past classes; set $\pi_c \propto \gamma\tilde{F}_c + (1 - \gamma)\widetilde{\Delta\ell}_c$

15:    **(D) Train classifier with mixed real and on the fly replay:**

16:    **for** classifier steps **do**

17:       sample a minibatch of size $B$: $B_r = \lfloor rB \rfloor$ real $(x, y) \sim \mathcal{D}_k$, $B_g = B - B_r$ synthetic

18:       sample class labels $\{c_j\}_{j=1}^{B_g} \sim \pi$; draw $\tilde{x}_j \sim q_\phi(\cdot \mid y = c_j)$

19:       form batch $\mathcal{B} = \{(x, y)\}_{B_r} \cup \{(\tilde{x}_j, c_j)\}_{B_g}$

20:       compute $\mathcal{L}_{\text{cls}} = \frac{1}{|\mathcal{B}|} \sum_{(x,y) \in \mathcal{B}} \text{CE}(f_\theta(x), y)$

21:       compute EWC penalty $\mathcal{L}_{\text{EWC}} = \frac{1}{2} \sum_i F_i^\star (\theta_i - \theta_i^\star)^2$

22:       update $\theta \leftarrow \theta - \eta\nabla_\theta(\mathcal{L}_{\text{cls}} + \lambda\mathcal{L}_{\text{EWC}})$

23:    **end for**

24:    **(E) Anchor for next task:** $\theta^\star \leftarrow \theta$,   $F^\star \leftarrow F^{(k)}$

25: **end for**

---

- **CheXpert** Irvin et al. (2019): A large-scale chest X-ray dataset with 14 labelled findings. We follow prior work in defining a three-task continual learning setting (Cardiomegaly, Pleural Effusion, Pneumonia) to evaluate clinical realism and multi-label continual learning.

- **Preprocessing:**
  - MedMNIST 2D tasks resized to $224 \times 224$ and normalised to $[0, 1]$.
  - MedMNIST 3D tasks cropped or resampled to $64 \times 64 \times 64$ voxel volumes and normalised channel-wise.
  - CheXpert images resized to $224 \times 224$, normalised to $[0, 1]$, and binarised into positive/negative labels per finding.

- **Splits:**
  - MedMNIST: Standard training/validation/test splits with 10% of training data reserved for validation.
  - CheXpert: We use the official training/validation split and evaluate in the three-task continual learning setting using sigmoid cross-entropy loss for the multi-label problem.

## F.3 MODEL ARCHITECTURE

- **Classifier (ViT, lightweight):** 4 encoder blocks; token dimension 64; 8 attention heads; GELU activations; pre-norm layers; a learnable [CLS] token with sinusoidal positional encodings; and a 2-layer MLP classification head. Patches are $16 \times 16$ for 2D inputs and $8 \times 8 \times 8$ for 3D volumes. Sigmoid outputs are used for CheXpert (multi-label), while softmax is used for MedMNIST tasks (multi-class).

Table 8: Hyperparameters and budget accounting for each benchmark. Replay storage is capped at 100 MB across all methods. For exemplar methods, raw images are stored; for generative replay, only the generator checkpoint is stored, with replay counted as if samples were materialised under the same cap.

| Dataset | $\lambda$ (EWC) | $\gamma$ (FSR) | Task orders | Memory accounting |
|---|---|---|---|---|
| MedMNIST-2D | 100 | 0.5 | default, size-asc., size-desc. | 100 MB (images / synthetic) |
| MedMNIST-3D | 200 | 0.5 | default, size-asc., size-desc. | 100 MB (images / synthetic) |
| CheXpert | 50 | 0.75 | default, size-asc., size-desc. | 100 MB (images / synthetic) |

- **Diffusion generator (unified class-conditional DDPM):** A single shared U-Net with FiLM-style label embeddings, cosine noise schedule, $T{=}1000$ steps, and an $\epsilon$-prediction objective. The U-Net has 4 downsampling and 4 upsampling blocks with channels [64, 128, 256, 512] and group normalisation. For 3D tasks, we extend this design with 3D convolutions while keeping the conditioning scheme consistent.

### F.4 Hyperparameter Settings

- **Optimizer:** AdamW with $\beta_1 = 0.9$, $\beta_2 = 0.999$, weight decay $1{\times}10^{-4}$.
- **Learning rates:** Classifier $3{\times}10^{-4}$; DDPM $1{\times}10^{-4}$.
- **Batch size:** 128 for 2D tasks; 16 for 3D tasks.
- **Epochs per task:** 30 for MedMNIST tasks; 5–10 for CheXpert.
- **EWC coefficient:** As in Table 8 (per dataset).
- **Replay:** On-the-fly sampling from the unified DDPM with replay ratio $r{=}0.5$ in each minibatch; no synthetic sets are stored.
- **Diffusion schedule:** Cosine schedule with 1000 denoising steps.
- **Regularisation:** Dropout 0.1 in the ViT classifier head.

### F.5 Compute Time and Energy

- **Training Time:** Each task required approximately 3–6 hours on a Colab Pro A100 GPU and 6–10 hours on a local RTX 2080 Ti GPU.
- **Energy Consumption:** The RTX 2080 Ti drew an estimated peak of $\sim$300W. For a typical 8-hour run, this corresponds to $\sim$2.4 kWh per task. Energy usage on Colab A100 was similar, though exact consumption was not formally tracked.

### F.6 Evaluation Metrics

We evaluate each method using three complementary metrics across tasks and benchmarks to comprehensively assess classification performance and forgetting:

**Accuracy (Acc).** The proportion of correctly predicted labels over the test set. Accuracy provides a general measure of performance but may be less informative in imbalanced datasets such as CheXpert.

**Area Under the ROC Curve (AUC).** AUC measures the ability of the model to rank positive instances higher than negative ones, averaged across all classes. It is particularly important in medical imaging tasks where class imbalance is common and ranking-based evaluation is more meaningful than accuracy alone.

**Forgetting (F).** We adopt the standard continual learning forgetting metric defined as:

$$F = \frac{1}{T-1} \sum_{t=1}^{T-1} \max_{l \leq T} a_{t,l} - a_{t,T}$$

where $a_{t,l}$ is the accuracy on task $t$ after training on task $l$, and $T$ is the total number of tasks. Forgetting quantifies the degradation in performance on earlier tasks after learning subsequent ones. Lower values indicate stronger retention of prior knowledge.

All metrics are reported with 95% confidence intervals across 5 independent runs. For CheXpert, we report macro-averaged metrics over the cumulative label set at each step. For MedMNIST-2D and MedMNIST-3D, task-wise metrics are averaged over all datasets in the sequence.

### F.7 TRAINING PROCEDURE REFERENCE

For completeness, the full training procedure with a *single* unified class-conditional DDPM, Fisher-Scheduled Replay, and EWC is given in Algorithm 1 (Section E). Replay is sampled on the fly; no synthetic datasets or per-task generators are stored.

## G EMPIRICAL ESTIMATION OF REPLAY DIVERGENCE

To quantify replay fidelity, we estimate the divergence between the real task distribution $\mathcal{D}_k$ and its DDPM-generated counterpart $\hat{\mathcal{D}}_k$. Feature embeddings are extracted using a pretrained ViT encoder, followed by kernel density estimation (KDE) in the latent space. We compute a symmetric KL divergence:

$$\hat{D}_{\mathrm{KL}} \;=\; \tfrac{1}{2}\Big[\mathrm{KL}(q_{\mathcal{D}_k} \,\|\, q_{\hat{\mathcal{D}}_k}) \;+\; \mathrm{KL}(q_{\hat{\mathcal{D}}_k} \,\|\, q_{\mathcal{D}_k})\Big],$$

where $q_{\mathcal{D}_k}$ and $q_{\hat{\mathcal{D}}_k}$ denote KDE approximations of the respective latent distributions. Lower values indicate closer alignment and higher replay fidelity.

Table 9: **Replay divergence estimates.** Symmetric KL divergence between real and replayed task distributions (lower is better), and corresponding classifier AUC achieved when training with replay.

| Task | Sym. KL ($\downarrow$) | Replay AUC ($\uparrow$) |
|---|---|---|
| BloodMNIST | $0.041 \pm 0.007$ | 0.84 |
| PathMNIST | $0.064 \pm 0.005$ | 0.88 |
| RetinaMNIST | $0.058 \pm 0.006$ | 0.82 |
| Adrenal3D | $0.073 \pm 0.010$ | 0.79 |
| CheXpert | $0.049 \pm 0.006$ | 0.86 |

## H UNIFIED CONDITIONAL DDPM ACROSS TASKS

We train a *single* class-conditional DDPM across all tasks (including CheXpert), using FiLM-style label embeddings for class conditioning. This amortised generator is used *only at training time* for on-the-fly replay and keeps inference exemplar-free, task-ID-free. In practice, a unified checkpoint reduces storage and training overhead by $\sim 45\%$ relative to maintaining per-task generators (comparable parameter counts; fewer checkpoints), with a small fidelity drop on complex 3D volumes and chest X-rays.

Table 10: Replay performance with *per-task* vs. *unified* DDPM. Unified replay remains competitive on 2D tasks, with minor degradation on 3D and chest X-ray. FID (lower is better) is computed on held-out validation splits.

| Task | Replay AUC (Per-Task) $\uparrow$ | Replay AUC (Unified) $\uparrow$ | FID (Unified) $\downarrow$ |
|---|---|---|---|
| BloodMNIST | 0.85 | 0.84 | 7.8 |
| PathMNIST | 0.88 | 0.86 | 9.4 |
| RetinaMNIST | 0.82 | 0.80 | 10.2 |
| Adrenal3D | 0.79 | 0.74 | 16.5 |
| CheXpert | 0.86 | 0.84 | 8.5 |

*Notes.* (1) The unified DDPM shares weights across tasks; no synthetic images are stored. (2) Conditioning is class-only; no task ID is required at sampling. (3) The $\sim 45\%$ saving reflects fewer generator checkpoints and lower optimizer/EMA state across tasks under the same 100 MB replay budget.

## I QUALITATIVE REPLAY SAMPLES

Figure 5 presents a visual comparison between replay samples generated by DDPMs (left) and VAEs (right) for three representative tasks: BloodMNIST (top block), Adrenal3D (middle block), and CheXpert (bottom block).

**BloodMNIST (2D):** DDPM-generated samples exhibit sharper cytoplasm boundaries, smoother gradients, and fewer artefacts, while VAE samples appear blurrier with reduced edge contrast.

**Adrenal3D (3D):** DDPM reconstructions better preserve anatomical contours and inter-slice coherence, whereas VAE outputs suffer from structural blurring and inconsistent voxel textures.

**CheXpert (X-rays):** DDPMs generate more realistic pulmonary structures, rib edges, and soft-tissue textures, while VAEs lose cardiothoracic detail and introduce noticeable blurring, limiting their utility for clinically relevant replay.

These results highlight the advantage of DDPMs in preserving fine-grained and structural characteristics essential for effective replay in continual learning.

## J TASK ORDER ROBUSTNESS

We assess robustness to task order by training on both the canonical and the reversed curricula. Table 11 reports final accuracy together with the absolute change ($\Delta$) and percentage drop from reversal. While DER++ and SPM lose about five percentage points relative, diffusion replay alone is more stable, and **EWC-DR (ours)** is minimally affected ($-0.7$ pp, $-0.90\%$), indicating that high-fidelity replay mitigates distributional shocks and Fisher anchoring reduces order-induced drift.

Table 11: Final accuracy under task reordering. $\Delta$ is Reversed $-$ Canonical (pp). Percent drop is (Canonical $-$ Reversed)/Canonical $\times$ 100. Lower drop is better.

| Method | Canonical | Reversed | $\Delta$ (pp) | Drop (%) |
|---|---|---|---|---|
| DER++ | 62.0 | 58.9 | $-3.1$ | 5.00 |
| SPM | 64.6 | 61.3 | $-3.3$ | 5.11 |
| Diffusion-only (DDPM) | 75.7 | 73.2 | $-2.5$ | 3.30 |
| **EWC-DR (ours)** | **78.2** | **77.5** | $-0.7$ | **0.90** |

## K REPLAY BUDGET SENSITIVITY

We evaluate accuracy as a function of replay buffer size. EWC-DR (ours) maintains strong performance down to 50 MB (Figure 6), unlike buffer-based methods that degrade below 100 MB.

Interpretation: diffusion replay yields high-entropy, class-consistent samples, enabling accurate rehearsal with far fewer stored points. This is promising for on-device continual learning under tight memory budgets.

## L LOW-SHOT GENERALISATION

We simulate data-constrained settings by reducing the per-task training set. Table 12 shows that EWC-DR (ours) outperforms all baselines at 10%, 25%, and 50% of the data.

Analysis: (i) diffusion replay acts as implicit data augmentation; (ii) EWC provides soft parameter anchoring that curbs overfitting in low-data regimes.

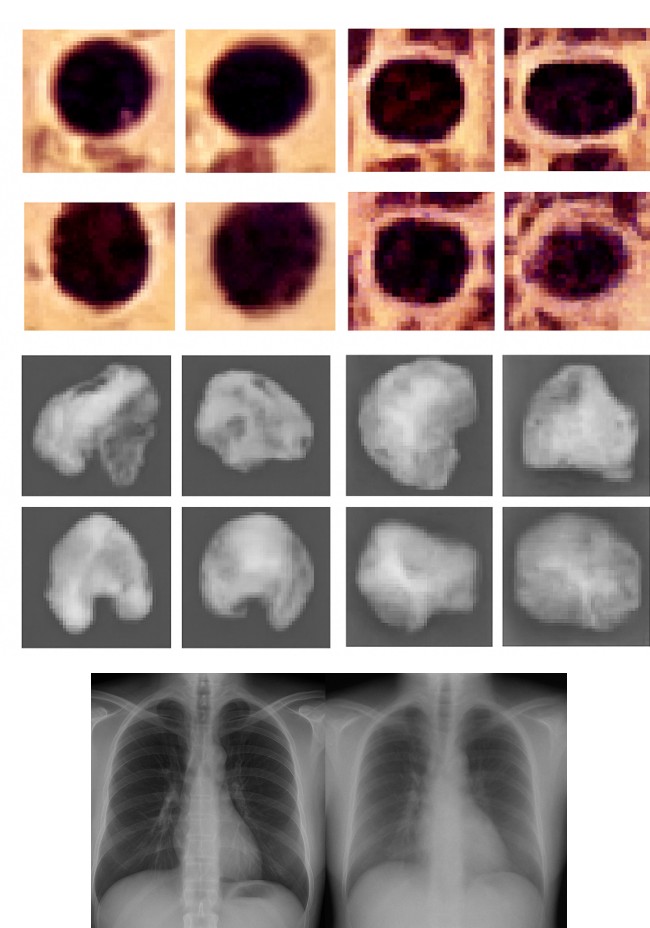

Figure 5: Side-by-side comparison of replay samples from DDPMs (left) and VAEs (right). **Top:** BloodMNIST. **Middle:** Adrenal3D. **Bottom:** CheXpert. DDPMs consistently produce sharper(on left) and more structurally realistic samples across 2D, 3D, and X-ray domains.

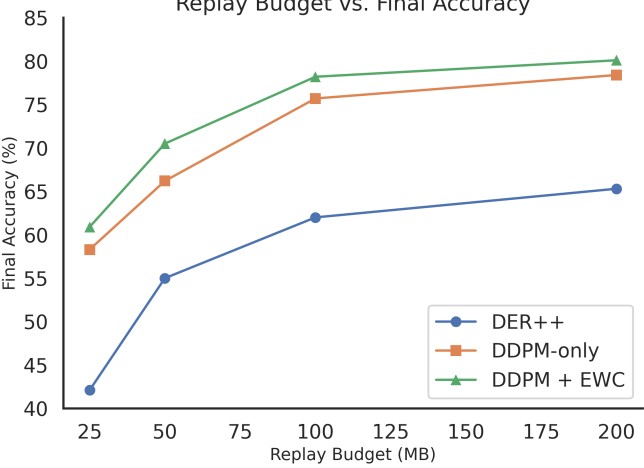

Figure 6: Final accuracy vs. replay buffer size. EWC-DR (ours) remains memory-efficient and robust at low budgets.

Table 12: Average accuracy with limited data per task.

| Method | 10% Data | 25% Data | 50% Data |
|---|---|---|---|
| DER++ | 38.5 | 51.2 | 60.3 |
| SPM | 40.1 | 53.8 | 62.5 |
| Diffusion-only (DDPM) | 51.6 | 65.7 | 74.3 |
| EWC-DR (ours) | **54.8** | **68.2** | **76.1** |

## M    ABLATION: DDPM GENERATION SETTINGS

To understand generation-performance trade-offs, we vary:

- **Timesteps** $T \in \{100, 250, 500, 1000\}$.
- **Noise Schedules** (linear, cosine).

As shown in Figure 7, cosine scheduling with $T \geq 500$ yields best results. Shorter $T$ speeds up sampling but harms fidelity.

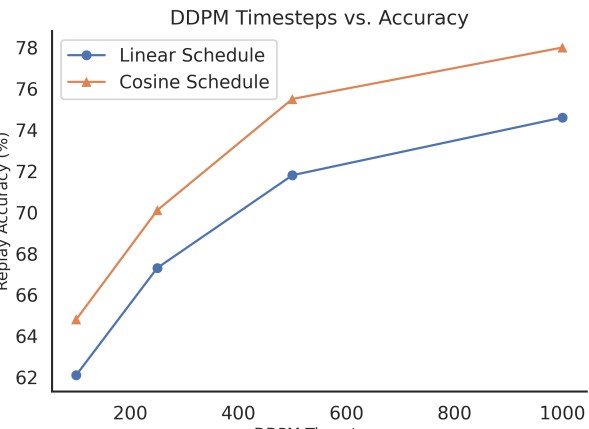

Figure 7: Replay accuracy under varying DDPM generation settings.

## N    KNOWLEDGE TRANSFER METRICS

We quantify forward transfer (FWT) and backward transfer (BWT) following standard continual-learning practice. Let $A_{i,j}$ be the test accuracy on task $j$ after training up to task $i$. Then

$$\text{FWT} = \frac{1}{T-1} \sum_{j=2}^{T} \big(A_{j-1,j} - A_{0,j}\big), \qquad \text{BWT} = \frac{1}{T-1} \sum_{j=1}^{T-1} \big(A_{T,j} - A_{j,j}\big).$$

FWT measures how prior learning helps new tasks (higher is better); BWT measures retention on past tasks after learning all $T$ (less negative is better).EWC-DR (ours) achieves positive FWT and near-zero (least-negative) BWT, indicating that diffusion replay aids forward generalisation while Fisher-anchored consolidation suppresses interference.

## O    LIMITATIONS AND FUTURE WORK

Our design choices aim to isolate core mechanisms while keeping the study tractable; we note their implications and planned extensions.

Table 13: Average forward transfer (FWT, higher is better) and backward transfer (BWT, closer to 0 is better).

| Method | FWT ↑ | BWT ↑ |
|---|---|---|
| DER++ | -0.017 | -0.093 |
| SPM | 0.003 | -0.071 |
| Diffusion-only (DDPM) | 0.045 | -0.031 |
| **EWC-DR (ours)** | **0.062** | **-0.017** |

Task design. We adopt a fixed canonical order with balanced replay to control confounders and attribute effects to replay and consolidation. This does not capture all clinical realities (e.g., class imbalance, evolving taxonomies, non-stationary curricula). We partially probed sensitivity via order reversal (Appendix J); next, we will evaluate imbalance-aware schedulers, dynamic curricula with distributional shocks, and open-world task discovery.

Datasets and external validity. MedMNIST v2 provides controlled 2D/3D tasks and CheXpert increases realism for radiography. These choices prioritise reproducibility and breadth over full resolution and multi-site heterogeneity. To strengthen external validity, we are preparing full-resolution studies across additional modalities (MRI, digital pathology) and multi-site cohorts, using privacy-preserving pipelines to respect governance constraints.

Efficiency and practicality. Diffusion replay adds training-time sampling cost, though inference remains generator-free and identical to the classifier. We chose standard DDPMs to establish a clear fidelity baseline. Future work will reduce cost via progressive distillation, few-step samplers, lightweight backbones, and cached class-conditioned priors, alongside reporting energy and latency under matched accuracy.

Baselines and evaluation protocol. We focus on exemplar-free and joint-training references under a unified memory policy (generator checkpoints counted toward the budget), to avoid privacy leakage. This omits hybrid exemplar and parameter-efficient continual adaptation (adapters/LoRA). We will expand comparisons to PEFT-based CL and privacy-preserving approximations to joint training, and include calibration and fairness metrics under class-imbalanced replay.

Theory and measurement. The bound assumes bounded loss and local quadratic behaviour with (diagonal) Fisher; divergence is estimated in latent space. These approximations make the bound conservative but interpretable and measurable. We plan tighter, task-adaptive constants, alternatives to diagonal Fisher, streaming divergence estimators, and sensitivity analyses over $\lambda$ and replay budget. Empirical correlations in Appendix D.5 already support the decomposition's predictive value.

