# OpenReview forum: "EWC-Guided Diffusion Replay for Exemplar-Free Continual Learning"
_ICLR.cc/2026/Conference — ICLR 2026 Conference Withdrawn Submission_

### Official Review · Reviewer_D8nW · 2025-10-29

**Soundness:** 1
**Presentation:** 2
**Contribution:** 1
**Rating:** 0
**Confidence:** 5

**Summary:**

This paper introduces a method for exemplar-free class-incremental learning in medical imaging that couples a single class-conditional diffusion model for generative replay with EWC for parameter anchoring, and adds Fisher-Scheduled Replay (FSR) to allocate synthetic samples to fragile classes. The paper motivates the design with a replay-vs-drift forgetting decomposition and reports gains over regularization and replay baselines on MedMNIST-2D/3D and CheXpert, using a shared ViT backbone, 100 MB budget, and 5-seed CIs. The paper falls short of top-tier exemplar-free CL standards in benchmarking scope, evaluation metrics, memory fairness, theoretical strength, and fidelity measurement. In its current form I recommend strong reject.

**Strengths:**

+ Well-done ablation study that include “EWC-only”, “DDPM-only”, “w/o FSR”, and DDGR; five-seed paired t-tests are reported.
+ A single shared conditional generator reduces parameters vs per-task generators; basic efficiency numbers (VRAM/params/train time/sampling) are tabulated.

**Weaknesses:**

+ This paper does not meet top EFCIL benchmarking standards. Evidence is limited to MedMNIST-2D/3D and a custom 3-task CheXpert. There is no evaluation on standard CL suites (e.g., CIFAR-100, CORe50, DomainNet, ImageNet variants), no task-order robustness study, and no cross-site transfer—well below the breadth expected in exemplar-free continual learning papers at top venues.
+ All methods are said to obey a 100 MB cap, but generative replay “stores only the generator checkpoint and samples on-the-fly,” while exemplar replay “stores raw images,” and replay storage is “counted as if synthetic samples were materialized.” This hybrid policy mixes storage and hypothetical sample bytes and risks under-counting the (large) generator; it is not aligned with common, strictly byte-accounted protocols.
+ The “forgetting bound” relies on smoothness, Pinsker-style arguments, and diagonal-Fisher local quadraticity; it functions as an explanatory lens but lacks tight constants, verification of assumptions, or actionable guarantees. The paper’s own empirical “validation” is correlational.
+ Replay fidelity measurement is unconvincing. Divergence is approximated via latent-space KDE/KL rather than more established generative metrics (FID/KID, precision–recall); conclusions about diffusion’s “lower drift” thus rest on a representation-dependent proxy.
+ I find baseline coverage and experimental fairness highly insufficient. While DDGR, DER++, SPM, CoPE, EFT, and a dynamic expansion baseline (PMoE) are included, there is no broader comparison to recent exemplar-free adapters/regularizers under matched memory and compute, nor to strong non-diffusion generative alternatives with identical schedules. The ablations do not isolate FSR against simpler schedulers under budget-matched sampling.
+ Metric breadth is below field norms. Top EFCIL papers report ACC, BWT/FWT, in-stream accuracy curves, and often calibration; here the focus is on final accuracy/AUROC and a scalar “forgetting,” with t-tests. Missing standard diagnostics hinder comparability
+ I find a big mismatch between the problem framing and the broad claims. The medical-only scope (small MedMNIST; bespoke CheXpert split) and the absence of multi-site evaluation make external validity unclear. Conclusions about “privacy-sensitive settings” are not backed by deployment-style studies.

**Questions:**

+ Can you add standard CL suites (CIFAR-100 class-inc. at minimum) and BWT/FWT + in-stream metrics to meet established EFCIL reporting norms?
+ Please replace or complement latent-KDE KL with FID/KID and/or precision–recall for generative models, and relate fidelity metrics to forgetting reductions.
+ Can you include budget- and schedule-matched ablations that isolate FSR from simpler allocators (uniform, class-prior, loss-only, Fisher-only) and from diffusion-only replay under identical sampling?

---

### Official Review · Reviewer_vKPg · 2025-10-30

**Soundness:** 2
**Presentation:** 4
**Contribution:** 2
**Rating:** 2
**Confidence:** 4

**Summary:**

This paper proposes EWC-guided Diffusion Replay, a hybrid continual learning method for medical imaging that operates without storing patient data. It combines a class-conditional diffusion model for generating synthetic data (replay) with Elastic Weight Consolidation for parameter stability. A key innovation is Fisher Scheduled Replay, which strategically allocates replays based on parameter importance and recent loss.

**Strengths:**

This paper proposes EWC-guided Diffusion Replay. It combines a class-conditional diffusion model for generating synthetic data (replay) with Elastic Weight Consolidation for parameter stability. A key innovation is Fisher Scheduled Replay, which strategically allocates replays based on parameter importance and recent loss. The method achieves competitive accuracy and reduced forgetting on medical image datasets, outperforming baselines.

**Weaknesses:**

The innovation of this paper is limited. While the work combines EWC with replay, its main contribution appears to be the proposed Fisher Scheduled Replay. However, the authors fail to provide an in-depth analysis or explanation of its underlying principles, and even the specific mechanism for allocating replay samples remains unclear.

Furthermore, the paper is poorly written, making it difficult to grasp the motivation and follow the logic. Significant redundancy exists between Sections 2 and 3, which could be consolidated. There is no dedicated related works section in the main text, it is only provided in the supplementary materials. Additionally, many equations in the paper are unnumbered.

The method uses a diffusion model to generate replay data, which in theory also requires continual learning. This critical point is not addressed in the main text and is only briefly mentioned in the supplementary materials. The effectiveness of the diffusion model's own continual learning directly impacts the overall results, and the authors should include relevant analysis in the experiments section, along with an evaluation of the method's computational efficiency.

There is also an error in the results labeling in Table 1, where the second-best performance is incorrectly marked.

**Questions:**

What is the specific underlying principle of Fisher Scheduled Replay, and how does it operate concretely in the replay process?
How does the "Theoretical Analysis of Forgetting" provided in the paper motivate Fisher Scheduled Replay? What is the motivation behind this?

---

### Official Review · Reviewer_asid · 2025-10-31

**Soundness:** 3
**Presentation:** 3
**Contribution:** 2
**Rating:** 4
**Confidence:** 3

**Summary:**

The authors propose EWC–Guided Diffusion Replay (EWC–DR) for exemplar-free, task-ID-free continual learning in medical imaging. The introduced method integrates:

(i) A single class-conditional diffusion generator amortized across tasks for high-fidelity replay,

(ii) Elastic Weight Consolidation (EWC) to anchor Fisher-salient parameters, and

(iii) Fisher-Scheduled Replay (FSR) that allocates synthetic samples to fragile classes via a convex combination of Fisher saliency and recent loss drift.

The authors present a forgetting decomposition that upper-bounds forgetting by a sum of replay divergence (KL between real and replayed data) and Fisher-weighted parameter drift, motivating the hybrid design. On MedMNIST v2 (2D/3D) and CheXpert under a strict 100 MB memory cap, EWC–DR outperforms strong regularization and replay baselines (DER++, SPM, VAE+Replay) and a diffusion-only replay baseline (DDGR) on accuracy/AUROC and average forgetting, with ablations validating each component and statistical tests (paired t-tests) indicating significance.

**Strengths:**

- Principled objective decomposition (design). The forgetting bound clearly motivates diffusion replay (reduce KL) + EWC (constrain Fisher drift) + FSR (budget where it matters). The paper validates how both terms correlate with forgetting and shows additive gains.

- Exemplar-free, task-ID-free CL with a single generator. Amortized class-conditional DDPM used only during training keeps inference light and privacy-friendly. It matched 100 MB budget ensures fair comparisons.

- Consistent cross-benchmark improvements. EWC–DR improves Acc/AUROC and reduces forgetting vs. regularization, exemplar replay, and DDGR under identical settings. The paired t-tests indicate significance.

- Solid engineering & reporting. Clear protocols, seeds, task-order robustness, ablations (w/o EWC, w/o FSR, diffusion-only), and efficiency analysis (VRAM/params/time/sampling).

**Weaknesses:**

- KL estimation and realism auditing. The KL term is approximated (latent KDE) and may correlate with aesthetics rather than diagnostic fidelity. A more per-class calibration/FID-like or feature-space distances could strengthen the link between replay quality and forgetting.

- Limited baseline diversity. Baselines omit some strong class-incremental methods (e.g., ER-ACE/DER w/ balanced rehearsal, GCR, or prompt/adapters) and non-diffusion generative baselines beyond VAE. Including at least one recent coreset/adapter method would contextualize gains.

- FSR specifics and stability. The scheduler mixes Fisher and loss-drift with a fixed γ. Shown sensitivity/robustness across streams, imbalance, and label-noise regimes is not deeply explored (CheXpert noise is mentioned, but targeted analyses would help).

- Privacy & compliance discussion is brief. While exemplar-free helps, generative replay may still reveal training patterns. A short discussion of membership-inference/attribute-inference risks or DP variants would anticipate deployment concerns in clinical settings.

**Questions:**

- FSR sensitivity: How sensitive are results to γ and the EMA parameters for loss-drift? Could the authors report a small grid (e.g.,  γ ∈ {0.25,0.5,0.75}, two EMA half-lives) and per-class replay counts vs. retention?

- Replay quality probes: Beyond downstream metrics, do the authors observe improved feature-space fidelity (e.g., Inception/MedNet features) or per-class calibration when using diffusion replay? Is there any failure cases where replay harms calibration?

- Adapters/prompts baseline: Could the authors add a lightweight adapter/prompt CL baseline (no exemplars) to clarify whether EWC–DR’s benefits persist when plasticity is routed through small modules?

- 3D scaling: For 3D DDPM, what is the sampling throughput vs. 2D? How does this affect budgeted replay ratios? Any instability or mode-dropping on NoduleMNIST3D?

- Bound calibration: Can the authors provide the joint regression coefficients a,b and R^2 per benchmark for the forgetting model
F_j = a * \hat(KL) + b * D_j + ϵ? This would make the decomposition’s predictive value more concrete.

**Details Of Ethics Concerns:**

N/A (public, de-identified datasets; exemplar-free training). I would recommend to consider a brief discussion about privacy auditing for generated replay and licensing/compliance of CheXpert/MedMNIST derivatives.

---

### Official Review · Reviewer_DBRb · 2025-11-03

**Soundness:** 2
**Presentation:** 1
**Contribution:** 1
**Rating:** 2
**Confidence:** 5

**Summary:**

The paper proposes EWC DR, a continual learning method that combines a single class conditional diffusion generator for replay with Elastic Weight Consolidation on the classifier. It also introduces Fisher Scheduled Replay, which allocates synthetic samples to classes using a mixture of per class Fisher saliency and recent validation loss drift. The analysis gives a forgetting decomposition that upper bounds average forgetting by a sum of a replay divergence term and a Fisher weighted parameter drift term, which matches the design of the method. Experiments on MedMNIST 2D, MedMNIST 3D, and CheXpert report higher final performance and lower forgetting than regularization and replay baselines under a fixed memory budget.

**Strengths:**

* The paper presents an interesting idea of generating targeted data based on the Fisher importance metric.

**Weaknesses:**

**Major Concerns:**

* The generator is pretrained on all future classes, which limits use when new classes appear. It cannot adapt without retraining the generator.
* Ablations are very limited. Each part of the method, such as diffusion replay, Fisher-based weighting, and the scheduling mechanism, could be replaced by different alternatives. However, the paper does not explore or compare these options. Section 6.5 only includes a written description of what happens when a component is removed, without showing quantitative results or a summary table. A more complete ablation study is needed to show the contribution and importance of each component.
* The method is general continual learning, yet all experiments are only on medical datasets.
* The consolidation relies on an older regularization idea while many newer methods exist, and the paper does not justify this choice.
* Benchmarks are small and MNIST like, which weakens the claims.
* The memory comparison appears unfair. Buffer-based and buffer-free methods are mixed, the 100 MB rule is vague, and what matters is the number of stored samples and buffer capacity. Experimental details are not sufficient to ensure fair comparisons.


**Minor Concerns:**

* The paper is unorganized. Method details and mathematical definitions appear after the experimental section, which makes it hard to follow.
* There is no related work section in the main paper, and the coverage of prior literature is too limited.

**Questions:**

* I would appreciate if the authors could clarify why the EWC regularization method was specifically chosen for consolidation instead of exploring more recent or advanced regularization-based continual learning techniques.
* It would be helpful if the authors could provide clearer experimental details on how the 100 MB budget is applied and explain more precisely how the evaluation protocol ensures fairness between replay-based and non-replay-based continual learning methods.

---

### Note · Authors · 2025-11-18

I have read and agree with the venue's withdrawal policy on behalf of myself and my co-authors.